# Cellular Uptake and Transport Mechanism Investigations of PEGylated Niosomes for Improving the Oral Delivery of Thymopentin

**DOI:** 10.3390/pharmaceutics16030397

**Published:** 2024-03-14

**Authors:** Mengyang Liu, Darren Svirskis, Thomas Proft, Jacelyn Loh, Yuan Huang, Jingyuan Wen

**Affiliations:** 1School of Pharmacy, Faculty of Medical and Health Sciences, The University of Auckland, 95 Park Road, Grafton, Auckland 1010, New Zealand; m.liu@auckland.ac.nz (M.L.); d.svirskis@auckland.ac.nz (D.S.); 2Department of Molecular Medicine and Pathology, Faculty of Medical and Health Sciences, The University of Auckland, Auckland 1010, New Zealand; t.proft@auckland.ac.nz (T.P.); mj.loh@auckland.ac.nz (J.L.); 3Maurice Wilkins Centre for Molecular Biodiscovery, The University of Auckland, Auckland 1010, New Zealand; 4Key Laboratory of Drug Targeting and Drug Delivery, West China School of Pharmacy, Sichuan University, Chengdu 610017, China; huangyuan@scu.edu.cn

**Keywords:** cellular uptake, cellular transport, niosome, PEGylation, thymopentin, oral delivery

## Abstract

Background: Although its immunomodulatory properties make thymopentin (TP5) appealing, its rapid metabolism and inactivation in the digestive system pose significant challenges for global scientists. PEGylated niosomal nanocarriers are hypothesized to improve the physicochemical stability of TP5, and to enhance its intestinal permeability for oral administration. Methods: TP5-loaded PEGylated niosomes were fabricated using the thin film hydration method. Co-cultured Caco-2 and HT29 cells with different ratios were screened as in vitro intestinal models. The cytotoxicity of TP5 and its formulations were evaluated using an MTT assay. The cellular uptake and transport studies were investigated in the absence or presence of variable inhibitors or enhancers, and their mechanisms were explored. Results and Discussion: All TP5 solutions and their niosomal formulations were nontoxic to Caco-2 and HT-29 cells. The uptake of TP5-PEG-niosomes by cells relied on active endocytosis, exhibiting dependence on time, energy, and concentration, which has the potential to significantly enhance its cellular uptake compared to TP5 in solution. Nevertheless, cellular transport rates were similar between TP5 in solution and its niosomal groups. The cellular transport of TP5 in solution was carried out mainly through MRP5 endocytosis and a passive pathway and effluxed by MRP5 transporters, while that of TP5-niosomes and TP5-PEG-niosomes was carried out through adsorptive- and clathrin-mediated endocytosis requiring energy. The permeability and transport rate was further enhanced when EDTA and sodium taurocholate were used as the penetration enhancers. Conclusions: This research has illustrated that PEG-niosomes were able to enhance the cellular uptake and maintain the cellular transport of TP5. This study also shows this formulation’s potential to serve as an effective carrier for improving the oral delivery of peptides.

## 1. Introduction

The peptide thymopentin (TP5) has recently attracted significant attention owing to its immunomodulatory properties and its potential application in the treatment of immunological diseases in both humans and animals [1,2]. TP5, illustrated in Figure 1, is a compact pentapeptide with therapeutic properties, consisting of the amino acids Arginine, Lysine, Aspartic Acid, Valine, and Tyrosine, and possessing a molecular weight of 679 g/mol. Initially extracted from the bovine thymus, it was later identified at the molecular level as a single-peptide chain named thymopoietin II, consisting of 49 amino acids, as depicted in Figure 1 [3]. Early in 1985, Rinaldi Garaci et al. revealed that TP5 is able to stimulate dendritic cells to provide resistance against fungal infections through the activation of antifungal T-cell helper 1 (Th1) responses, with the involvement of Toll-like receptor (TLR) 2 signaling being its potential immunomodulatory mechanism [4]. TP5 has shown to have pleiotropic effects in immunomodulation including T cell and B cell differentiation, IgG production, T cell proliferation, and Interleukin-2 production [5]. Therefore, TP5 has been investigated for its potential efficacy in treating ailments such as severe acute respiratory syndrome (SARS) by ameliorating clinical symptoms, enhancing immune function recovery, reducing lung inflammation, minimizing hormone dosage, and shortening treatment duration [6].

Similar to other peptide and protein medications, the only currently accessible form of TP5 is a lyophilized powder designed for intramuscular or intravenous administration [3]. This limitation is attributed to the instability and rapid metabolism (degradation) experienced when administered through alternative routes. Several factors contribute to the inadequate oral absorption of TP5, such as high degradation in the gastrointestinal (GI) tract, a low log P, and a high molecular weight [7,8,9]. Transdermal and alternative non-invasive administration routes pose challenges for the absorption of TP5, including issues related to low skin or tissue permeability [10,11,12,13,14]. The physicochemical properties of TP5 limit its efficacy as an immunomodulator for immune-related diseases in both humans and animals [15]. Employing formulation strategies may be useful to surmount the inherent constraints of TP5, thereby facilitating its oral delivery [16,17].

PEGylation, which involves the attachment of PEGs to a protein or nucleic acid molecule, can increase its aqueous solubility, prolong its circulation time, protect it from in vivo biological inactivation via proteolysis, and reduce its immunogenicity in some cases [18]. The primary purpose of using PEGylation technology to improve drug stability in vivo is to prolong the half-life of the drug, which can reduce its clearance via the kidneys [19]. PEGylating a drug delivery system like niosome can improve systemic circulation time and decrease immunogenicity to obtain higher stability [20]. Niosomes offer several advantages in oral delivery, including possessing a (1) nanometer-sized particle to enhance the bioactive absorption; (2) a spherical shape to expand contact area for drug absorption; (3) an encapsulation structure to improve the stability of drug; and (4) a controlled and sustained drug release profile to reduce the frequency of drug administration [20].

The human adenocarcinoma cell lines Caco-2 and HT-29 have been developed as a co-cultured model for the intestinal epithelium. Caco-2 and HT-29 cells were originally isolated from the human colon adenocarcinoma by Fogh et al. [21]. The co-cultured Caco-2 and HT-29 cell system is commonly employed in an in vitro intestinal absorption model in both industry and academia for assessing the permeability of various compounds in the intestine [21,22]. These cells also showcase the presence of nutrient and drug transporters, enabling the investigations of carrier-mediated uptake and efflux mechanisms [21,22,23]. 

In this study, a series of in vitro cell assays were used to explore the oral delivery of TP5. In specific, the cytotoxicity of both niosomes and TP5 were investigated. The cellular uptake studies were also assessed for TP5 with and without niosome and PEGylation technologies. Various absorption inhibitors and enhancers were employed to investigate the transport mechanisms of TP5-PEG-niosomes, which may be affected by interactions between the drugs or formulations and intestinal cellular transporters. The main aim of this project is to illustrate enhanced cellular uptake and gain insights into the transport of TP5-PEG-niosomes, to achieve an enhance the oral delivery of TP5. This research contributes to the field by providing insights into the novel approach of employing TP5-PEG-niosomes for improved oral delivery, potentially offering beneficial therapeutic outcomes for TP5-based treatments.

## 2. Materials and Methods

### 2.1. Materials

TP5 (≥95% LC-MS validated) was purchased from Wuhan DanGang Biological Technology company (DGpeptide, Wuhan, China). Soybean trypsin and chymotrypsin inhibitors (SBTCI), PEG600-cholesterol, cholesterol (CH), stearic acid, sorbitan monostearate (Span^®^ 60), polyoxyethylenesorbitan monooleate (Tween^®^ 80), and dihexadecyl phosphate (DCP) were obtained from Merck (Merck, Darmstadt, Germany). Methanol and acetonitrile (ACN) of analytical reagent grade were ordered from Sigma-Aldrich (Sigma, San Jose, CA, USA). Phosphoric acid (PPA), trifluoroacetic acid (TFA), and ethylenediaminetetraacetic acid (EDTA) were obtained from Honeywell Fluka (Honeywell Fluka, Seelze, Germany). Dulbecco’s Modified Eagle Medium (DMEM) with high glucose and L-glutamine, Hank’s Balanced Salt Solution (HBSS), phosphate-buffered saline (PBS), non-essential amino acid solution (NEAA), heat-inactivated Fetal Bovine Serum (FBS), penicillin-streptomycin, and trypsin-EDTA were obtained from Invitrogen (Auckland, New Zealand). 3-(4,5-dimethylthiazol-2-yl)-2,5-diphenyltetrazolium bromide (MTT) was purchased from Sigma-Aldrich (Sigma, San Jose, CA, USA). All other reagents were of analytical grade.

### 2.2. Analytical Method of TP5

The RP-HPLC analytical method for TP5 was developed in a prior study [10]. In brief, TP5 peak separation was accomplished using a Gemini C18 column (250 × 4.60 mm, 5 µm; Phenomenex, Torrance, CA, USA) equipped with a C18 guard column (10 × 3.0 mm; Phenomenex, Torrance, CA, USA) and maintained at 25 ± 1 °C. The RP-HPLC-UV conditions included a wavelength of 210 nm, a flow rate of 1.0 mL/min, and a mobile phase consisting of 0.1% phosphoric acid (pH 2.6)/methanol with a ratio of 80:20 (*v*/*v*).

### 2.3. In Vitro Cell Culture Investigation

#### 2.3.1. Cell Co-Culture

Caco-2 or HT-29 cells were cultured in DMEM media, which is composed of 10% FBS, 1% non-essential amino acid (NEAA) and 1% penicillin-streptomycin (antibiotics) with pH values maintained at 7.4 ± 0.2. The cells were incubated in an incubator at 37 °C in the presence of 5% CO_2_. Caco-2 or HT-29 cells were fed fresh DMEM every 3 days until the confluence value reached 80% for the subculture. In brief, the DMEM medium was removed, and the Caco-2 or HT-29 cells were washed with pre-warmed PBS (37 °C) to remove traces of the serum, which may inhibit the action of trypsin. These cells were subsequently incubated with a 0.25% trypsin-EDTA (0.2 mL/cm^2^) mixture for about 3 min to guarantee all the cells were floating and detached. Later, an equal amount of FBS was added to the cell culture to deactivate trypsin. The Caco-2 or HT-29 cell suspension was then centrifuged at 300× *g* for 10 min. The supernatant was discarded, and the cells were re-suspended in the fresh DMEM medium. The secondary suspension was then divided into two parts and cultured separately using DMEM medium. The newly cultured Caoco-2 and HT-29 cells were used as co-cultured models in the following sections. 

#### 2.3.2. Cell Line Integrity

The Caco-2 and HT-29 cells were co-cultured in ratios of 1:0, 0:1, 1:1, 1:2, 1:3, 3:1, 5:1, 7:1, and 9:1. This co-cultured cell line was evaluated by transepithelial electrical resistance (TEER) values to determine the tissue integrity. The TEER measurements were performed at 3-day intervals until the confluence reached 80%. TEER = (Rm − Rb) × A, where Rm, Rb, and A were the measured TEER values, TEER value of the blank transwell, and the membrane surface area of the transwell, respectively. The ratio with the best TEER value was selected as the final ratio for further experiments.

#### 2.3.3. Preparation of TP5 Stock Solution and TP5-PEG-Niosomes

TP5 stock solution was prepared by dissolving TP5 in a cell culture medium, resulting in a final concentration of 10 mg/mL. TP5-PEG-niosomes were prepared and fabricated in two parts based on our previous research with particle size at 156 ± 36 nm [24]. In brief, it started with the conjugation of PEG600 and cholesterol. In the second part, TP5-PEG-niosomes were fabricated using a thin film hydration method [24]. In short, a 10 mL blend of methanol and chloroform (1:4, *v*/*v*) comprising 2 μmoL DCP, 30 µmol cholesterol (including 22.5 µmol PEG600-cholesterol), and 120 µmol Span 60 was prepared. The mixture was rotary evaporated (Laborota 4000, Buchi, Switzerland) at 45 °C to generate a thin, dry lipid film. This film was subsequently purified under a nitrogen flow until the residual organic solvent was eliminated. The final lipid film was hydrated with 10 mL cell DMEM medium, containing 10 mg of TP5, at 60 °C for 60 min to produce the TP5-PEG-niosomes. The TP5 concentration in the final formulation was prepared as 1 mg/mL. 

#### 2.3.4. Cytotoxicity Study

MTT assay was used for the cytotoxicity study. The Caco-2 and HT-29 co-cultured cell lines were cultured in 96-well plates (BD Falcon^TM^, Auckland, New Zealand) at a seeding density of 5000 cells per well for 24 h at 37 °C to promote cell adhesion. The DMEM was discarded, and the Caco-2 and HT-29 co-cultured cells were washed with PBS before introducing TP5 test samples. The cells were then treated with filtered TP5 solutions (0.005 to 10 mg/mL) or TP5-PEG-niosomes (niosome numbers 0.01 to 10.00 × 10^12^ /mL) and cultured for 24, 48, and 72 h. The cells were examined using a microscope (EvosTM XL Core microscope with a magnification of 10× (eyepiece lens) and 10× (objective lens) from Thermo Fisher Scientific^®^ (Thermo Fisher, Waltham, MA, USA). Subsequently, 20 μL of MTT (5 mg/mL) was added, and the cells were incubated for an additional 4 h. Dimethyl sulfoxide (DMSO) was introduced to dissolve the formazan crystals, and the absorbance at 570 nm was measured using a microplate UV-Vis spectrophotometer (SpectraMax^®^ Plus, Molecular Devices, San Jose, CA, USA). The percentage of cell viability was calculated in comparison to the untreated control group.

For each test sample group and at every time point over the course of three days, cell viability was computed. The outcomes were consolidated using the SoftMax Pro v5 software system through Equation (1).
(1)$$V=\frac{A}{A_{0}}\times 100\%$$
where V is the cell viability, A is sample absorbance, and A_0_ is control absorbance.

#### 2.3.5. Cellular Uptake

##### Fluorescence and Confocal Microscopy Analysis

First, TP5 was labeled with fluorescein isothiocyanate (FITC) [25]. Briefly, 25 mg of TP5 and 2.5 mg of FITC were dissolved in 5 mL of 0.1 M carbonate buffer, pH 8.5, and stirred in the dark for 8 h at 4 °C. The mixtures underwent fractionation through gel permeation chromatography utilizing a Sephadex G-25M column (Sigma Aldrich, St. Louis, MO, USA). In a concise procedure, 1 mL of the mixture was loaded onto the top of the column gel bed, and the column was eluted with 10 mL of PBS (0.01, pH 7.4). Fractions of 1 mL each were collected, and the absorbance of each fraction was assessed at 280 nm. During elution, two bands were discernible. The conjugates were identified in the first band (fractions 3–5, with absorbance > 0.4). Fractionated FITC-TP5 conjugates were dehydrated by freeze drying (Labconco, Kansas City, MO, USA), and loaded into PEG-niosomes as described previously.

Caco-2 and HT-29 co-cultured cells were transferred into 8-well chamber slides (BD Falcon, Fort Pierce, FL, USA) at a density of 10^4^ cells/cm^2^ and grown in a complete DMEM culture medium. On the following day (24 h culture), cell monolayers were pre-incubated with 1 mL of HBSS for 15 min at 37 °C. After equilibrating, the medium was replaced with 1 mL suspensions of FITC-TP5, FITC-TP5-loaded PEG-niosomes (1 mg TP5 per mL in HBSS) followed by incubation for 1 h at 37 °C (513). Then, the cells were washed eight times with cold sterile PBS (7.4) and fixed with freshly prepared 4% (weight per volume) (*w*/*v*) p- formaldehyde (PFA, pH 7.4) for 20 min, followed by cell nuclei staining with DAPI (100 nm in PBS) for 10 min. Sample time points were set as 0.5, 1, 2, and 3 h for the TP5 solution, and 0.5, 1, 2, 3, 4 and 5 h for the TP5 formulation. After removing the culture chambers, the slides were thoroughly washed with PBS and mounted with CITI-Fluor (a medium used to decrease photobleaching during observation under microscopy). Coverslips were placed on top of the slides and sealed with nail polish. 

Prior to the experiment, the slides were stored at 4 °C and protected from the light. The slides were visualized by fluorescence microscope (Nikon TE2000E, Nikon, Tokyo, Japan) first before using the laser scanning confocal microscope. Cell nuclear was stained with 0.1% Hoechst 33342 in 0.01M PBS containing 0.1% tween (PBST) for 15 min. To prevent photo-bleaching, a drop of mounting solution was added to the specimens before observation using a Zeiss™ LSM 710 inverted confocal microscope (Carl Zeiss Ltd., München, Germany). The excitation wavelength was set at 405 nm for Hoechst, 560 nm for Nile red and 488 nm for FITC-TP5. The samples were also investigated using a laser scanning confocal microscope (Leica TCS SP2, Bio-strategy, Rosedale, NZ, USA). 

The fluorescence images were evaluated with Leica Confocal Software (LCS) version 2.61 (Bio-strategy, Rosedale, NZ) and ImageJ™ software, version 1.5 (National Institutes of Health, Bethesda, MD, USA).

##### Quantitative Analysis of the Cellular Uptake of TP5 and TP5-PEG-Niosomes

Briefly, 5 mL of Caco-2 and HT-29 co-cultured cells’ suspension at a density of 10^5^ cells/cm^2^ was seeded onto 60 mm plastic dishes (Corning, New York, NY, USA), fed complete DMEM every 3 days and incubated at 37 °C in an atmosphere of 5% CO_2_ and 95% relative humidity. On reaching 80% confluence, the culture medium was replaced with HBSS (2 mL). After 30 min of incubation at 37 °C, the medium was replaced with 1 mL TP5 solution with different concentrations (5, 10, 50, 100, 200, 500, and 1000 µg/mL in HBSS) and TP5-PEG-niosomes (1 mg TP5 in 0.01, 0.05, 0.10, 1.00, 2.00, and 5.00 × 10^12^ numbers niosomes per mL in HBSS) and incubated for 4 h at 37 °C to investigate the effect of drug and niosome concentration on TP5 cellular uptake. For time-dependent cellular uptake experiments, the cells were incubated for 0.5, 1, 2, 3, 4 and 5 h at 37 °C, respectively. Then, the cells were washed with cold sterile PBS (pH 7.4) eight times and solubilized in 1 mL 10% Triton X-100 in methanol followed by the extraction of TP5. Then, 25 µL of the cell lysates from each well were subjected to a BCA protein assay (Thermo Scientific, Waltham, MA, USA) to determine the amount of cell protein. The entire protein concentration in each group was calculated based on the linearity equation obtained from the BSA standard curve. Then, cell lysates were centrifuged for 15 min at 14,000 rpm, and the supernatants were analyzed for the amount of uptake TP5 using the RP-HPLC method mentioned earlier. The uptake of TP5 by Caco-2 and HT-29 co-cultured cells was calculated and expressed as the amount of drug (ng) uptake per µg cell protein.

##### Cellular Uptake Mechanisms of TP5 and TP5-PEG-Niosomes

One of the mechanisms for cellular uptake depends on concentration. To carry out a drug-concentration-dependent cellular uptake study, aliquots of 2 mL TP5 of different concentrations (10, 50, 100, 200, 500 and 1000 μg/mL) in HBSS were added into the dish and incubated for 60 min in triplicate. To carry out a noisome-concentration-dependent cellular uptake study, 2 mL TP5-PEG-niosomes of different niosome numbers (0.01, 0.05, 0.10, 1.00, 2.00, and 5.00 × 10^12^ /mL measured by NanoSight NS300 (Malvern Instruments Ltd., Worcestershire, UK) and prepared by series dilution) in HBSS was added into the dish and incubated for 4 h in triplicate. The treated Caco-2 and HT-29 co-cultured cells were then scraped off into a 2 mL microcentrifuge tube containing 1.0 mL extraction solution (10% Triton X-100) and sonicated to accelerate cell lysis. Then, 25 µL of the cell lysates from each well were subjected to a BCA protein assay (Thermo Scientific, Waltham, MA, USA) to determine the amount of cell protein as mentioned above. Then, cell lysates were centrifuged for 15 min at 14,000 rpm, and the supernatants were analyzed for the amount of uptake TP5 using the RP-HPLC method mentioned above. The uptake of TP5 by Caco-2 and HT-29 co-cultured cells was calculated and expressed as the amount of drug (ng) uptake per µg cell protein.

To investigate other cellular uptake mechanisms, Caco-2 and HT-29 co-cultured cell monolayers were treated with different inhibitors and different conditions for 1 h before cellular uptake started. The inhibitors used in this study include an active transport inhibitor (sodium azide, 100 mM), an adsorptive-mediated endocytosis inhibitor (protamine sulphate, 1 mM), a clathrin-mediated endocytosis inhibitor (chlorpromazine, 10 μg/mL), and a caveolae-mediated endocytosis inhibitor (filipin, 5 μg/mL) [23]. The incubation temperature of 4 °C was also tested. After the removal of the drug solution and formulation (4 h incubation), the dishes were immediately rinsed twice with ice-cold HBSS solution (5 mL). The treated Caco-2 and HT-29 co-cultured cells were then scraped off into a 2 mL microcentrifuge tube containing 1.0 mL extraction solution (10% Triton X-100) and sonicated to accelerate cell lysis. Furthermore, 25 µL of the cell lysates from each well were subjected to a BCA protein assay (Thermo Scientific, Waltham, MA, USA) to determine the amount of cell protein mentioned above. Then, cell lysates were centrifuged for 15 min at 14,000 rpm, and the supernatants were analyzed for the amount of cellular uptake TP5 using the RP-HPLC method mentioned above. The uptake of TP5 by Caco-2 and HT-29 co-cultured cells was calculated and expressed as the amount of drug (ng) uptake per µg cell protein.

#### 2.3.6. Cellular Transport

##### Transwell Culture

Caco-2 and HT-29 co-cultured cells at a density of 10^5^ cells/cm^2^ were seeded on 12-well Insert Transwell^®^ inserts (0.4 µm pore diameter, 1.1 cm^2^ area, Corning, New York, NY, USA). Caco-2 and HT-29 co-cultured cells were incubated at 37 °C in an atmosphere of 5% CO_2_ and 95% relative humidity in complete DMEM. The medium was replaced every 3 days with 0.5 mL medium on the apical (AP) side and 1.5 mL on the basolateral (BL) side, as shown in Figure 2. The integrity of the co-cultured cell monolayer was assessed by monitoring transepithelial electrical resistance (TEER). TEER measurements were conducted using a Millicell-ERS Volt-Ohm meter connected to a pair of chopstick electrodes (Millipore Corp, Boston, MA, USA). The electrodes, sterilized with 70% ethanol for 15 min and air-dried for 15 s, were immersed in a manner where the shorter electrode was on the AP side, and the more extended electrode was on the BL side, ensuring they did not come into contact with the cell monolayer. TEER measurements were taken at 3-day intervals until the TEER values surpassed 350 Ω·cm^2^ (around 24 days post-seeding in this study). The TEER calculation utilized the formula TEER = (Rm − Rb) × A, where Rm, Rb, and A represent the measured TEER values, TEER values of the blank transwell, and the membrane surface area of the transwell (1.13 cm^2^), respectively. Caco-2 monolayers with TEER values exceeding 350 Ω·cm^2^ were deemed to have acceptable integrity and were considered suitable for the transport study.

##### Transepithelial Transport of TP5 and TP5-PEG-Niosomes

Before the transport experiments, Caco-2 and HT-29 co-cultured cell monolayers underwent equilibration with HBSS for 30 min and were subsequently aspirated. All transport studies were carried out at 37 °C. A 0.5 mL solution of TP5 and TP5-PEG-niosomes (0.1 mg TP5 per mL in HBSS) was introduced on the AP side, while the BL side of the inserts contained 1.5 mL of HBSS, as depicted in Figure 2. At 0, 60, 120, 240, and 360 min of incubation, 200 µL was withdrawn from the BL-receiving chamber and immediately replaced with an equal volume of pre-warmed HBSS (37 °C). The concentration of TP5 in the transport medium was analyzed using RP-HPLC.

##### Cellular Transport Mechanism of TP5 and TP5-PEG-Niosomes

TP5 transport mechanisms were investigated in the absence or presence of variable inhibitors or enhancers. As shown in Figure 2, there were numerous transporters in Caco-2 cell transport. By considering both apical and basolateral sides, active transport inhibitor (10 to 20 mM sodium azide, Adenosine triphosphate (ATP) inhibitor), clathrin-mediated endocytosis inhibitor (10 to 20 μg/mL chlorpromazine), permeability glycoprotein (P-gp) inhibitor (50 to 100 µM verapamil), multidrug-resistance-associated protein 2 (MRP2) inhibitor (100–150 µM MK-571), multidrug-resistance-associated protein 5 (MRP5) inhibitor (20–40 µM Silymarin), tight junction opener (10–20 mM Sodium taurocholate) and absorption enhancer (5–10 mM EDTA) were selected to investigate the cellular transport mechanism, as shown in Table 1.

The concentration of these selected inhibitors and enhancers was screened by cytotoxicity using Caco-2 and HT-29 co-cultured cells by MTT assay. Then, these inhibitors and enhancers with safe concentrations were added to the AP side with a 0.5 mL 1 mg/mL solution of TP5 and TP5-PEG-niosomes, respectively. All inhibitors were dissolved in dimethyl sulfoxide and diluted with HBSS. The transport characteristics of TP5 and TP5-PEG-niosomes through Caco-2 and HT-29 co-cultured cell monolayers were expressed as the cumulative transported drug and transport rate. The transport rate (Flux) was denoted in μg/min/cm^2^ and computed using the formula Flux = (dM/dt)/A, where dM/dt represents the cumulative amount of TP5 in the BL side per unit time (µg/min), and A is the surface area of the insert membrane (1.1 cm^2^). The apparent permeability (Papp) coefficient was expressed in cm/s and calculated as P*app* = (dM/dt)/(60 × A × C), where C represents the initial concentration (1 mg/mL). The TEER values of the monolayers were assessed at 1 h intervals throughout the experimental duration, guaranteeing the integrity of the monolayers.

### 2.4. Data Analysis

Statistical analysis of in vitro cell studies was carried out using GraphPad Prism^®^ version 8.0 software, while significance was determined using two-way ANOVA (*p* value < 0.05).

## 3. Results and Discussion

### 3.1. Analytical Method of TP5

Using the previously established analytical method to quantify the amount of TP5 in cell assays, a distinct peak with an adequate retention time was achieved [10]. The calibration curve, with an r^2^ value exceeding 0.999, was established, and the corresponding standard equation (Equation (2)) applicable to the working concentration range of 5 to 100 µg/mL is provided below:(2)Y=12.48X+0.2054(Y represents the area under the curve of the HPLC chromatogram, and X signifies the concentration of the drug. This equation is utilized for the quantification of TP5 in subsequent investigations).

### 3.2. In Vitro Cell Culture Investigation

#### 3.2.1. Cell Line Integrity

Caco-2 and HT-29 cells co-cultured at ratios of 1:0, 0:1, 1:1, 1:2, 1:3, 3:1, 5:1, 7:1, and 9:1 all displayed similar growth characteristics, reaching 80% confluence in x days (Figure 3). Layer integrity was assessed by TEER measurements every 3 days, and only TEER values exceeding 350 Ω·cm^2^ were considered suitable for further study [35,36,37]. The TEER values of newly cultured Caco-2 and HT-29 cells in different ratios are shown in Table 2. It can be seen that Caco-2 and HT-29 cells with ratios of 1:0, 7:1, and 9:1 met the requirement of tissue integrity. Because of the benefits of the co-culture of Caco-2 and HT-29 cells, it provides the presence of both absorptive and goblet cells, both of which have different culture requirements for optimal growth and function to mimic intestine tissue in vivo [38,39,40,41,42]. Therefore, the ratio of 9:1 was selected based on the presence of both cell types and highest TEER value for subsequent uptake and transport studies.

#### 3.2.2. Cytotoxicity Study

Cytotoxicity studies were conducted to investigate the effect of drug and formulation composition on the cell viability of Caco-2 and HT-29 cells in the presence of each TP5 (1 mg/mL) formulation (TP5 solution and TP5-PEG-niosomes) and their respective controls for 72 h (Figure 4). It can be seen that the morphology of Caco-2 and HT-29 cells was similar in all conditions, indicating less cytotoxicity of TP5 formulations [43,44]. However, a quantitative measure was required, in which the MTT assay was also used to determine the cell viability with a wide range of concentrations of the drug and formulation.

As shown in Figure 5a, TP5 was not toxic at all in the tested concentrations. As the concentration increased, there was an increase in proliferation until reaching 5.000 mg/mL, followed by similar cell viability when the concentration reached 10.000 mg/mL. This indicates that non-cytotoxicity can be observed in all TP5 solutions at three different time intervals (24, 48, and 72 h). This is in correspondence with the results from the literature that TP5 is able to promote cell growth because TP5 can provide nutrition to Caco-2 and HT-29 cells due to its amino acid nature [43,44]. Cytotoxicity studies were also conducted to investigate the effect of formulation concentration on the cell viability of Caco-2 and HT-29 cells in the presence of TP5. Caco-2 and HT-29 cells were incubated with PEG-TP5-niosome of niosome numbers ranging from 0.01 to 10.00 × 10^12^ /mL for 24, 48, and 72 h, respectively. As shown in Figure 5b, PEG-niosomes were not toxic in most concentrations from 0.01 to 2.00 × 10^12^ /mL for 24, 48, and 72 h. As the niosome numbers increased, there was a light increase in cell growth until the niosomes’ number reached 2.000 × 10^12^ /mL, followed by a decreased cell viability after the concentration reached 5.00 × 10^12^ /mL. It may be due to the weak toxicity of surfactants used in PEG-niosomes [45,46]. Because the higher numbers of niosomes contain a higher drug loading, the cellular uptake and transport of TP5 can be increased. Therefore, PEG-niosomes with a concentration under 2.00 × 10^12^ /mL provide a safety profile when treated on human Caco-2 and HT-29 cells according to the increased cell viability compared to the blank control group.

#### 3.2.3. Cellular Uptake

##### Fluorescence and Confocal Microscopy Analysis

TP5 was labeled with fluorescein isothiocyanate (FITC) [25]. The cellular uptake of FITC-TP5 by Caco-2 and HT-29 co-cultured cells after 0.5, 1, 2, and 3 h was visualized by fluorescence and confocal microscopy. As shown in Figure 6 and Figure 7, the green fluorescence of FITC-TP5 was observed within cells after 0.5 h incubation at 37 °C, and continued to increase in cells until at least 3 h. Overlaying images of cell nuclei (blue) and cytoplasm (red) indicated that TP5 was found in the cytoplasm.

The cellular uptake of FITC-TP5-PEG-niosomes by Caco-2 and HT-29 co-cultured cells was explored over a more extended period because of the controlled release profile of TP5 from PEG-niosomes. The cellular uptake at 0.5, 1, 2, 3, 4, and 5 h was visualized by fluorescence microscopy first and then by confocal microscopy analysis. As shown in Figure 8 and Figure 9, the green fluorescence of FITC-TP5-PEG-niosomes was observed within cells after 0.5 h incubation at 37 °C in contrast to cell nuclei (blue) and cytoplasm (red), which, indicating TP5 from FITC-TP5-PEG-niosomes, were internalized into cells within 0.5 h and reached the highest concentration at 3 to 5 h because of the highest fluorescence intensity observed at this time point.

It can be seen that the resolution and pixels improved when confocal laser scanning microscopy was used compared to fluorescence microscopy. Therefore, the intensity data from confocal laser scanning microscopy were selected to be analyzed by ImageJ™ software, version 1.5 (National Institutes of Health, Bethesda, MD, USA). As shown in Figure 10a, the cellular uptakes of FITC-TP5 solution and FITC-TP5-PEG-niosomes suspension were quantified by their fluorescence intensity obtained from the data of confocal laser scanning microscopy. The pure drug exhibited the highest intensity of cellular uptake at 2 h, whilst that of the niosomal formulation was observed at 4 h. This confirmed the controlled release profile of the designed formulation. Furthermore, there was also no significant difference after 3 h, indicating the optimized formulation has drug absorption in the intestine similar to that of a pure drug. This behavior emphasized the possibility of high intestinal permeability and effective drug absorption in the human digestive system [47].

##### Quantitative Analysis of the Cellular Uptake of TP5 and TP5-PEG-Niosomes

The cellular uptake was further investigated quantitatively by HPLC and BCA assay (Figure 10b). These results corresponded to the fluorescence-intensity-derived data, in which TP5 exhibited the highest amount of cellular uptake drug at 2 h, whilst that of the niosomal formulation was observed at 3 to 5 h. This further confirmed the time-dependent uptake of TP5 and TP5-PEG-niosomes by Caco-2 and HT-29 co-cultured cells. TP5 reached saturation/equilibrium after 2 h, whereas TP5-PEG-niosome uptake was slower. In detail, for pure drug solution, the cellular uptake value was increased to 3.56 ± 1.13 ng/µg in 2 h and maintained at this amount for the remaining hours. In the group of TP5-PEG-niosomes with the equivalent amount of 1 mg/mL TP5, the cellular uptake value steadily increased to 4.05 ± 0.94 ng/µg, and slightly increased to 4.58 ± 1.21 ng/µg. This longer cellular uptake of TP5-PEG-niosomes indicates the long circulating ability to reach the target area and therefore enhances the therapeutic effect on site [48].

##### Cellular Uptake Mechanisms of TP5 and TP5-PEG-Niosomes

As shown in Figure 11, the cellular uptake of TP5 was drug concentration and noisome concentration dependent from up to 1000 µg/mL TP5 and 5.00 × 10^12^ /mL the number of niosomes. Specifically, the TP5 cellular uptake value can reach 41.32 ± 10.33 ng/µg maximum when a dose of 1 mg/mL TP5 is given, whereas the TP5 cellular uptake value can be improved to 6.87 ± 1.23 ng/µg maximum when there are higher numbers of niosomes used for drug delivery. However, considering the safety issues (cytotoxicity studies of PEG-niosomes), a niosome number lower than 2.00 × 10^12^ /mL should be used for TP5 oral delivery.

To investigate whether active cellular uptake mechanisms were involved for TP5 and its formulation, inhibitors that block adsorptive-mediated endocytosis, clathrin-mediated endocytosis and caveolae-mediated endocytosis or a reduction in temperature were used. The results are shown in Figure 12.

The cellular uptake amount of TP5 was significantly reduced by using 1 mM protamine sulphate, in which the adsorptive-mediated endocytosis pathway was blocked (*p* < 0.005). This indicated that TP5 cellular uptake occurred primarily through adsorptive-mediated endocytosis. Interestingly, when the treatment was changed to a temperature of 4 °C, the cellular uptake amount of TP5 was also significantly decreased. It can be hypothesized that the cellular uptake of the TP5 solution is partially energy dependent, and both passive and active pathways were available for the pure drug of TP5. The results also suggested that TP5-PEG-niosomes were primarily taken up via the caveolae-mediated endocytosis pathways because the cellular uptake values significantly decreased when the pathways were blocked by 5 µg/mL filipin. Cellular uptake was also partially blocked by 10 µg/mL Chlorpromazine, indicating the involvement of clathrin-mediated endocytosis. This corresponds to the results of other studies in which lipid-based nanocarriers mainly follow caveolae- or clathrin-mediated endocytosis for cellular uptake in intestinal epithelial membranes [49,50,51,52].

In summary, the cellular uptake of the TP5 solution by Caco-2 and HT-29 co-cultured cells is concentration, time, and temperature dependent, and mainly follows adsorptive-mediated endocytosis and a partial passive pathway. The cellular uptake mechanism of TP5-PEG-niosomes is also concentration, time, and temperature dependent, but follows caveolae- or clathrin-mediated endocytosis pathways.

#### 3.2.4. Cellular Transport

##### Transwell Culture and Cell Line Integrity

Caco-2 and HT-29 were co-cultured at a density of 10^5^ cells/cm^2^ onto 12-Well Insert Transwell^®^ inserts (0.4 µm pore diameter, 1.1 cm^2^ area, Corning, New York, NY, USA) with a ratio of 9:1. In order to determine the integrity of cell monolayers, the TEER measurements were performed at daily intervals for 3 days until the TEER values exceeded 350 Ω·cm^2^ (after ~24 days post seeding in this study). As shown in Table 3, Caco-2 and HT-29 monolayers displayed the TEER values 408.2 ± 36.5 Ω cm^2^ at testing day 1, 466.8 ± 29.8 Ω cm^2^ at testing day 2, and 485.7 ± 66.8 Ω cm^2^ at testing day 3, which were all above the 350 Ω·cm^2^ required to meet intestinal integrity for transport studies [35,36,37].

##### Transepithelial Transport of TP5 and TP5-PEG-Niosomes

Prior to the cellular transport experiments, cumulative amount, flux, and the apparent permeability coefficient (P*app*) values were investigated for pure drug solution, traditional niosome without PEGylation, and PEG-niosomes. As shown in Figure 13a, there is no significant difference in the cumulative amount of TP5 amongst the TP5 solution, TP5-niosomes, and TP5-PEG-niosomes. This indicates that TP5 niosomal formulation could reach the same cellular transport effect as the drug solution. The result of flux is shown in Figure 13b, in which all three groups reached the highest speed of 30 × 10^−4^ μg/min/cm^2^ at 120 min, and the speed became stable for 6 h. As listed in Table 4, there is also no difference in P*app* values between the TP5 solution and TP5-niosomes or TP5-PEG-niosomes. All three groups displayed high P*app* values around 5 × 10^−8^ cm/s for transport compared with other peptides such as insulin [53,54,55,56]. This also emphasizes that TP5 formulations have a similar apparent permeability coefficient compared with the drug itself. Employing niosomal technology ensures equivalent drug transport rates as free drug solutions while safeguarding the drug against enzymatic degradation. This is attributed to the small molecular weight of TP5, which facilitates easier traversal across epithelial membranes, alongside its delicate fragile nature.

##### Cellular Transport Mechanism of TP5 and TP5-PEG-Niosomes

The effects of absorption inhibitors or enhancers on transport were investigated in this experiment to explore the TP5 cellular transport. In brief, active transport inhibitor (10–20 mM sodium azide), clathrin-mediated endocytosis inhibitor (10 to 20 μg/mL chlorpromazine), permeability glycoprotein (P-gp) inhibitor (50 to 100 µM verapamil), multidrug-resistance-associated protein 2 (MRP2) inhibitor (100–150 µM MK-571), multidrug-resistance-associated protein 5 (MRP5) inhibitor (20–40 µM Silymarin), tight junction opener (10–20 mM Sodium taurocholate), and absorption enhancer (5–10 mM EDTA) were selected to investigate cellular transport mechanism. However, the safety of these treatments should be investigated. The cytotoxicity of these inhibitors, transporters, and enhancers was evaluated before the transport studies. The result of cytotoxicity is shown in Figure 14a. It can be seen that the safe but still effective range concentrations for treatments were 20 mM sodium azide, 100 µM verapamil, 10 μg/mL chlorpromazine, 100 µM MK-571, 20 µM Silymarin, 10 mM EDTA, and 20 mM Sodium taurocholate. These concentrations were used for the cellular transport studies to investigate the cellular transport mechanism of TP5, TP5-niosomes, and TP5-PEG-niosomes.

The influence of absorption enhancers or inhibitors on the cumulative amount of TP5 drug solution is shown in Figure 14b. At the first 60 min, the TP5 cumulative amount was displayed similarly in all treatment groups. After that, both parameters tended to increase sharply in the EDTA and sodium taurocholate groups and a slight increase was observed in other groups. This indicates that EDTA and sodium taurocholate can work as transport enhancers to improve the TP5 cellular transport. Compared to the control group (drug solution only), when ATP inhibitor (20 mM sodium azide), MRP2 inhibitor (100 µM MK-571), and MRP5 inhibitor (silymarin) were added, the drug cumulative amount of TP5 was reduced. This indicates that TP5 cellular transport partially requires energy, and TP5 may have its own transporters or other peptide transporters on the apical and basolateral membranes of intestinal epithelial cells.

Compared to the drug solution, niosome displayed a lower cumulative amount and cellular transport results in the treatment of sodium azide and verapamil, as shown in Figure 15a. This indicates the cellular transport of niosome was based on caveolae-mediated endocytosis with the energy required. Similar to the drug solution, the cellular transport rate was significantly increased when EDTA and sodium taurocholate were used, showing a 2.8-fold and 2.4-fold enhancement, respectively.

The influence of absorption enhancers or inhibitors on the cumulative amount of the TP5-PEG-niosome formulation was shown in Figure 15b. Compared to the drug solution, PEGylated niosome displayed a reduced cumulative amount and transport in the treatment of sodium azide, silymarin, and verapamil. This indicates the cellular transport of niosome was still based on caveolae-mediated endocytosis with the energy required. However, there is no significant difference amongst drug solution, niosomes, and PEG-niosomes without any treatments, which shows that the niosome formulation can reach the same cellular transport rate as the TP5 solution. Similar to previous findings, the TP5 cumulative amount can be enhanced by 2.6-fold and 2.1-fold when EDTA and sodium taurocholate are used, respectively.

The P*app* values of TP5 in all conditions are shown in Table 5. It can be seen that the rank order of P*app* values of TP5 solution is EDTA > sodium taurocholate >> verapamil > sodium azide > chlorpromazine > no treatment > MK571 > silymarin, and the values range from 4.30 ± 0.55 × 10^−8^ to 13.36 ± 0.52 × 10^−8^ cm/s. This shows that the main cellular transport mechanism of the pure drug solution is partially via multidrug-resistance-associated protein 5 transporter, an active endocytosis, and partially via passive transport. The rank order of P*app* values of TP5-niosomes without PEGylation is EDTA > sodium taurocholate >> no treatment > MK571 > verapamil > chlorpromazine > silymarin > sodium azide, and the values range from 3.15 ± 0.41 × 10^−8^ to 10.73 ± 0.77 × 10^−8^ cm/s. This indicates that active cellular transport and clathrin-mediated endocytosis are the main cellular transport mechanisms, and using EDTA can largely improve drug transport efficacy. The rank order of P*app* values of TP5-PEG-niosomes is EDTA > sodium taurocholate >> MK571 > no treatment > chlorpromazine > silymarin > verapamil > sodium azide, and the values range from 2.76 ± 0.18 × 10^−8^ to 9.05 ± 0.64 × 10^−8^ cm/s. Because of the controlled release profile, the P*app* values in PEG-niosome group were reduced in most treatments compared to traditional niosomes and drug solutions. The lowest Papp value of 2.76 ± 0.18 × 10^−8^ cm/s in the sodium azide group emphasized that the PEG-niosome cellular transport required energy and may follow clathrin-mediated endocytosis for cellular uptake and cellular transport.

The transporters, namely clathrin-mediated transporter, P-gp, MRP2, and MRP5, are predominantly present on the apical and basolateral membranes of the intestinal epithelia [57,58,59,60]. These transporters play essential roles in impeding the absorption of drugs by facilitating their efflux from the gut epithelial cells back into the intestinal lumen, thus preventing their uptake into the bloodstream [61,62,63]. Hence, inhibiting the efflux pump could potentially augment the cellular transport of drugs into the cells. The primary mechanism leading to an increased concentration of TP5 in cellular transport studies was observed with the inhibition of MRP5. These results, in conjunction with research conducted by others, indicate that MRP5 predominantly influences the regulation of plasma concentrations for various peptide and protein drugs [64]. These results indicate that the penetration across Caco-2 and HT-29 monolayer was energy dependent, and an active cellular transport mechanism was involved. Compared with blank control (drug solution without niosomal formulations), the addition of P-gp inhibitor (100 μM verapamil) or active transport inhibitor (20 Mm sodium azide) caused a significant decrease in the Papp of PEG-niosomes from 5.39 ± 0.46 × 10^−8^ cm/s to 3.06 ± 0.16 × 10^−8^ cm/s and 2.76 ± 0.18× 10^−8^ cm/s, respectively. This suggested that P-gp might play a significant role in the cellular transport of TP5-niosomes and TP5-PEG-niosomes on the Caco-2 and HT-29 cell membranes and showed energy dependence. The P*app* values from AP to BL direction in all groups across Caco-2 and HT-29 monolayers exhibited a notable increase in the presence of sodium taurocholate or EDTA. This effect is typically presumed to impact tight junctions between adjacent epithelial cells by modulating paracellular permeability [65]. The process of enhancing the absorption of lipid particles may be elucidated by potentially reducing the extracellular calcium concentration (via the addition of EDTA) or by opening tight junctions of cell monolayers (in the presence of sodium taurocholate) [66,67,68]. This could result in increased fluidity of cell membranes, disturbance in protein structure across the membranes, and enhancement of the endocytosis pathway.

## 4. Conclusions

In this article, in vitro studies were conducted to investigate the cell cytotoxicity, cellular uptake, and cellular transport of TP5 solution, TP5-niosomes, and TP5-PEG-niosomes. This study showed a significant improvement regarding using a formulation strategy compared with a pure drug solution towards the cellular uptake and transport of TP5. The cytotoxicity of TP5 pure solution, TP5-niosomes, and TP5-PEG-niosomes on the growth of Caco-2 and HT-29 co-cultured cell lines was investigated using MTT assay. None of the three formulations showed any toxic effects (TP5 concentration: 0 to 10 mg/mL, and niosome numbers: 0 to 2.00 × 10^12^ /mL) on intestinal Caco-2 and HT-29 cells over three days. The cellular uptake of TP5-PEG-niosomes was based on active endocytosis, and was time, energy, and concentration dependent, which could remarkably improve the cell uptake of TP5. The cellular transport rate was similar in the TP5 solution, TP5-niosomes, and TP5-PEG-niosomes, indicating the formulation has no effect on the drug cellular transport rate. The cellular transport of TP5 occurs mainly through MRP5 endocytosis and a passive pathway and is effluxed by MRP5 transporters, while that of TP5-niosomes and TP5-PEG-niosomes is carried out through adsorptive- and clathrin-mediated endocytosis requiring energy. The permeability and cellular transport rate can be further enhanced when EDTA and sodium taurocholate are used as the penetration enhancers. Overall, the data obtained highlight the safety and feasibility of using PEGylated niosomes as a drug delivery system. This innovative formulation is expected to improve the stability of TP5 and enhance its permeability across intestinal epithelial membranes. Therefore, further studies are warranted to investigate its in vivo efficacy when administered orally for the treatment of immunodeficiency.

## Figures and Tables

**Figure 1 pharmaceutics-16-00397-f001:**
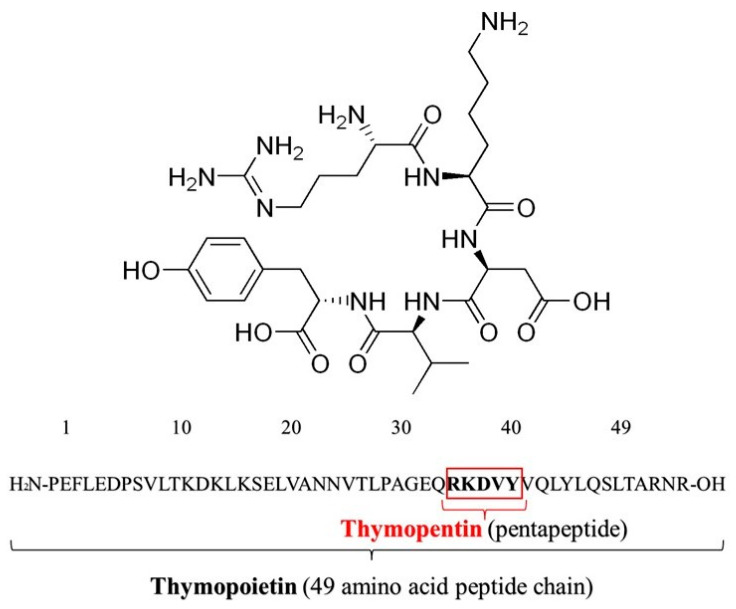
The chemical structure of TP5 and the amino acid sequences of thymopoietin and thymopentin.

**Figure 2 pharmaceutics-16-00397-f002:**
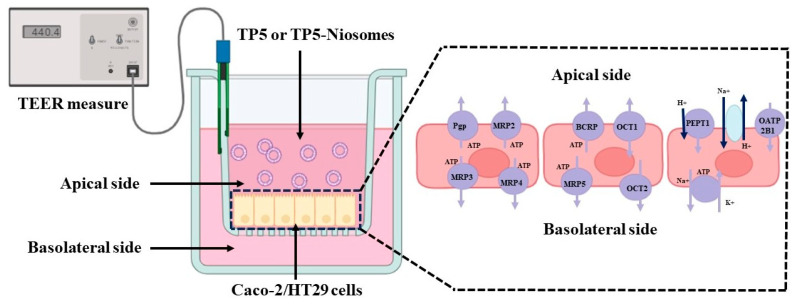
Schematic diagrams of transcellular transport and different transporters in Caco-2 and HT-29 co-cultured cell monolayers.

**Figure 3 pharmaceutics-16-00397-f003:**
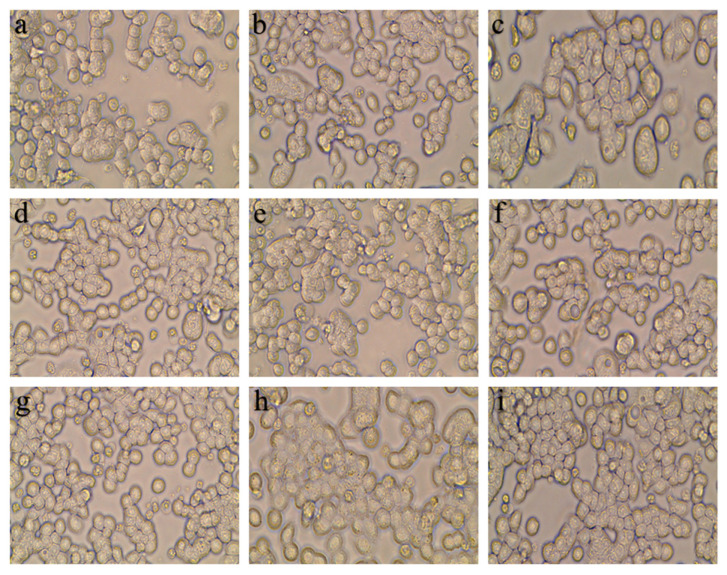
The micrographs of Caco-2 and HT-29 cells after 80% confluence with ratios of (**a**) 1:0, (**b**) 0:1, (**c**) 1:1, (**d**) 1:2, (**e**) 1:3, (**f**) 3:1, (**g**) 5:1, (**h**) 7:1, and (**i**) 9:1.

**Figure 4 pharmaceutics-16-00397-f004:**
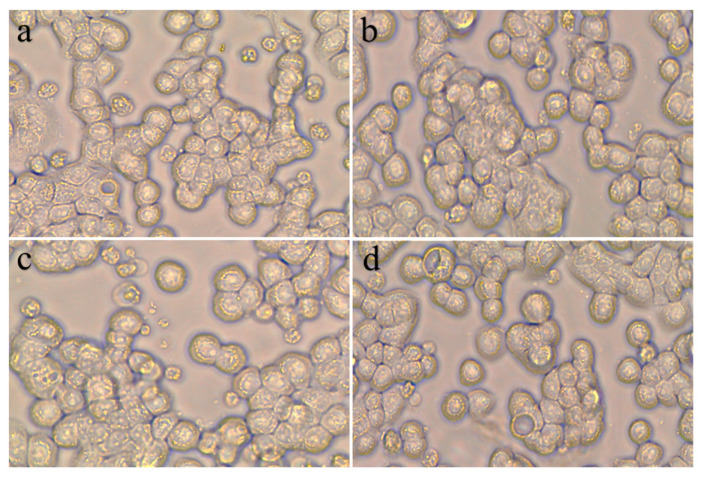
The micrographs of Caco-2 and HT-29 cells after 72 h, (**a**) Caco-2 and HT-29 cells without any TP5 formulations; (**b**) Caco-2 and HT-29 cells treated with TP5 solution; (**c**) Caco-2 and HT-29 cells treated with blank PEG-niosomes suspension; (**d**) Caco-2 and HT-29 cells treated with TP5-PEG-niosomes.

**Figure 5 pharmaceutics-16-00397-f005:**
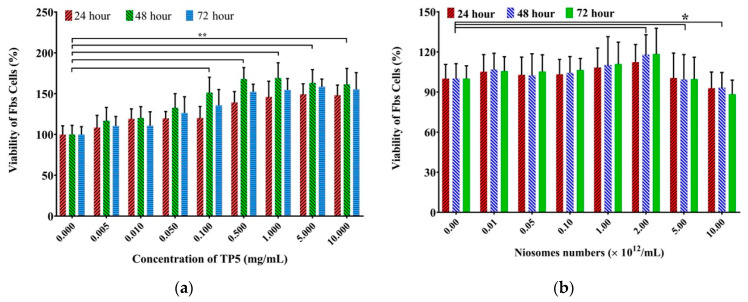
MTT viability assay showing the effect of TP5 formulations on co-cultured Caco-2 and HT-29 cells. (**a**) Effect of increasing TP5 concentrations in solution; (**b**) effect of increasing PEG-TP5-niosome concentrations. Cells were incubated with drug at 37 °C for 24, 48, and 72 h (*p* value ˂ 0.05 * and 0.01 ** to control) (Mean ± S.D., *n* = 6).

**Figure 6 pharmaceutics-16-00397-f006:**
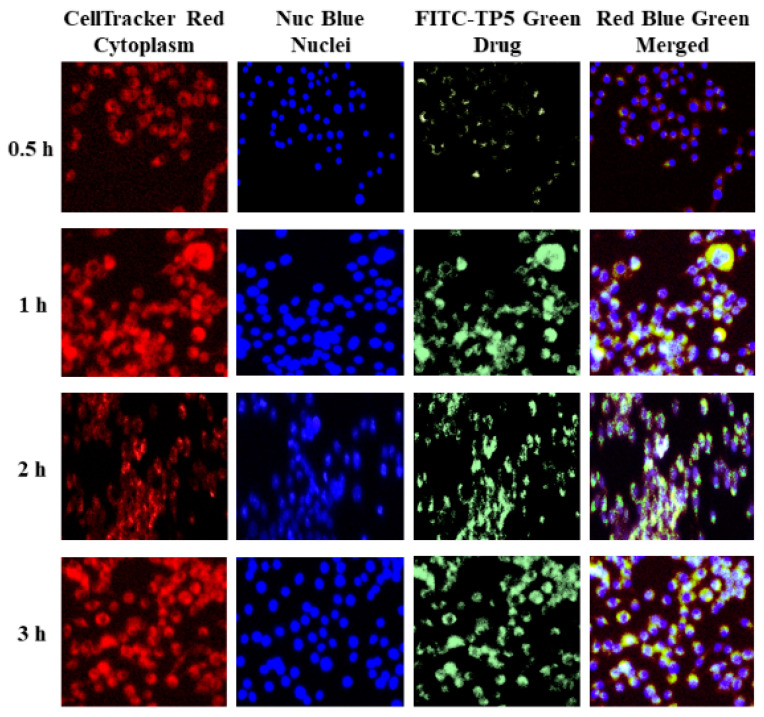
Fluorescence microscopy images of Caco-2 and HT-29 cells treated with FITC-TP5 (green) at 0.5, 1, 2, and 3 h in contrast to cell nuclei (blue), cytoplasm (red), and merged (green, blue, and red) (magnification = 500×).

**Figure 7 pharmaceutics-16-00397-f007:**
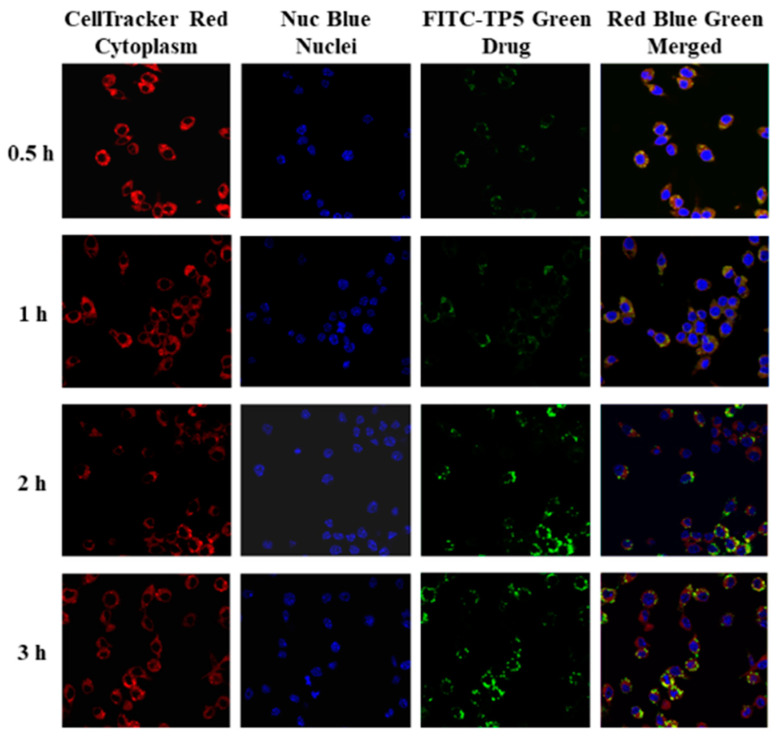
Confocal laser scanning microscopy 3D cross-section images of Caco-2 and HT-29 cells treated with FITC-TP5 (green) at 0.5, 1, 2, and 3 h in contrast to cell nuclei (blue), cytoplasm (red), and merged (green, blue, and red) (magnification = 500×).

**Figure 8 pharmaceutics-16-00397-f008:**
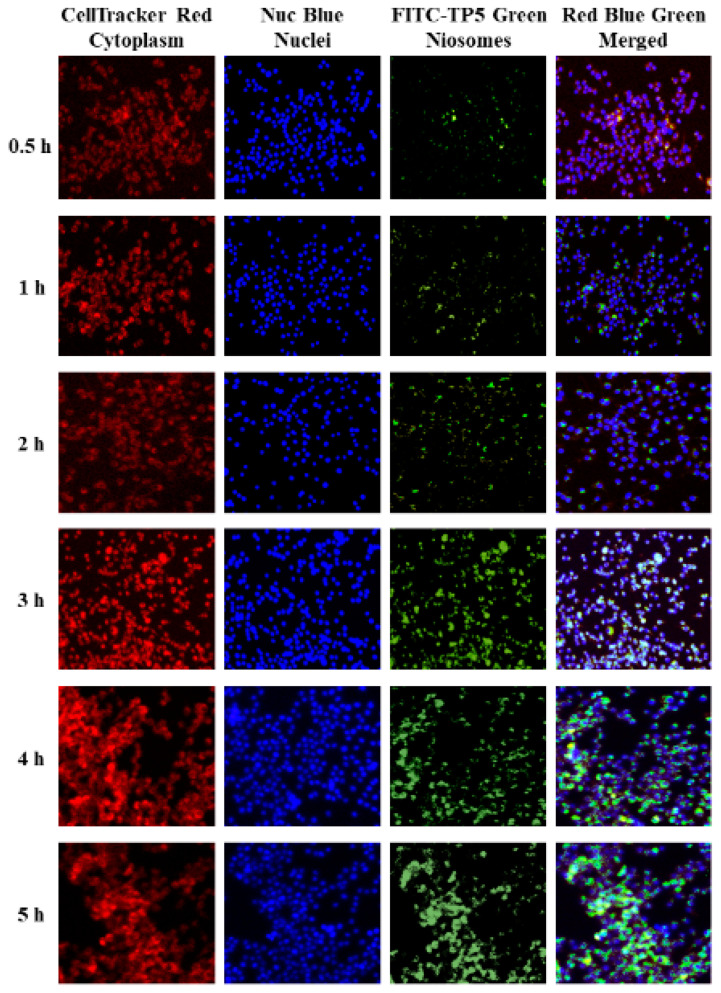
Fluorescence microscopy images of Caco-2 and HT-29 cells treated with FITC-TP5-PEG-niosomes (green) at 0.5, 1, 2, 3, 4, and 5 h in contrast to cell nuclei (blue), cytoplasm (red), and merged (green, blue, and red) (magnification = 500×).

**Figure 9 pharmaceutics-16-00397-f009:**
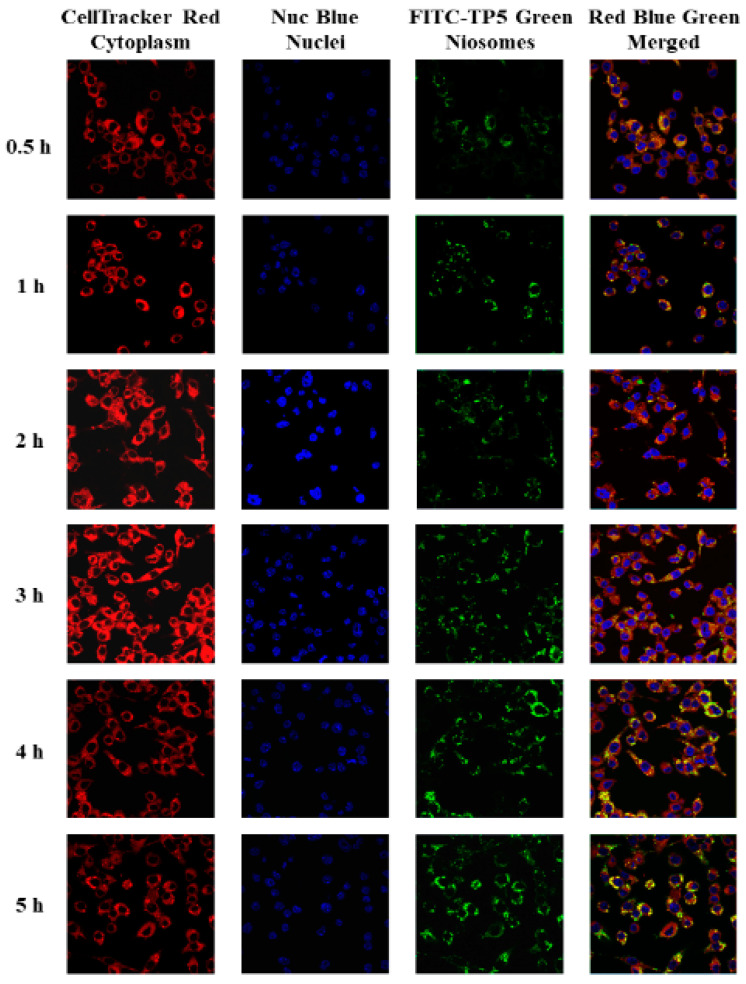
Confocal laser scanning microscopy 3D cross-section images of Caco-2 and HT-29 cells treated with FITC-TP5-PEG-niosomes (green) at 0.5, 1, 2, 3, 4, and 5 h in contrast to cell nuclei (blue), cytoplasm (red), and merged (green, blue, and red) (magnification = 500×).

**Figure 10 pharmaceutics-16-00397-f010:**
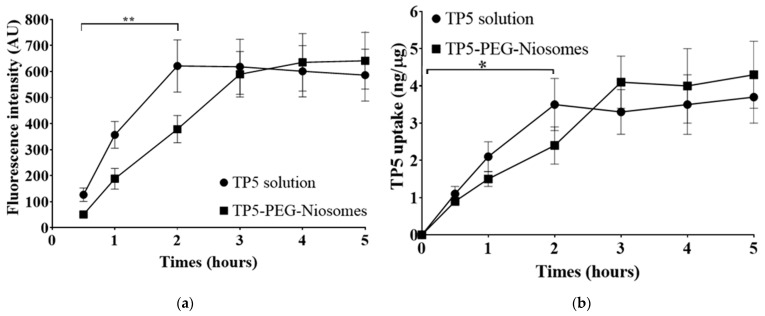
(**a**) Fluorescence-intensity-derived cellular uptake of Caco-2 and HT-29 cells treated with FITC-TP5-PEG-niosomes (green) at 0.5, 1, 2, 3, 4, and 5 h; (**b**) HPLC and BCA quantitative cellular uptake of Caco-2 and HT-29 cells treated with FITC-TP5-PEG-niosomes (green) at 0.5, 1, 2, 3, 4, and 5 h (Mean ± S.D., *n* = 3) (0.05 * and 0.01 ** < 0.05).

**Figure 11 pharmaceutics-16-00397-f011:**
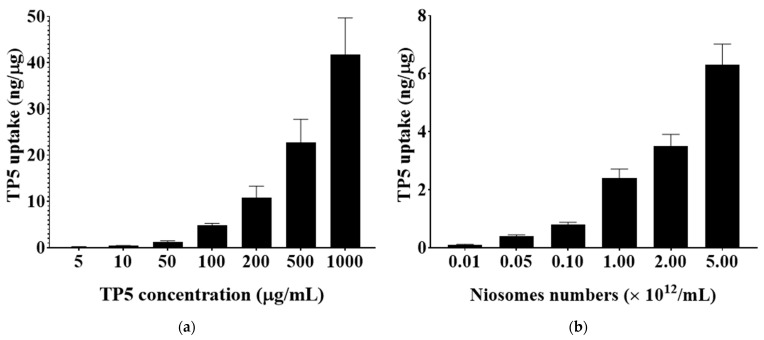
(**a**) The effect of TP5 concentration on TP5 cellular uptake values; (**b**) the effect of niosomes’ concentration on TP5 cellular uptake values (mean ± S.D., *n* = 5).

**Figure 12 pharmaceutics-16-00397-f012:**
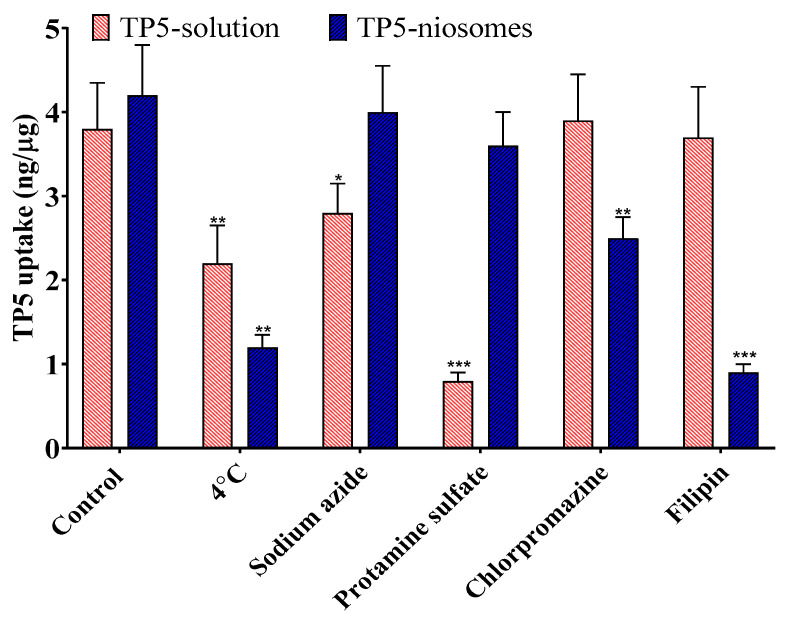
The effect of inhibitors and low temperature on TP5 and its formulation cellular uptake for 4 h on Caco-2 and HT-29 co-cultured cell monolayers (Mean ± S.D., *n* = 5; *** *p* < 0.005, ** *p* < 0.01, * *p* < 0.05 comparing to control).

**Figure 13 pharmaceutics-16-00397-f013:**
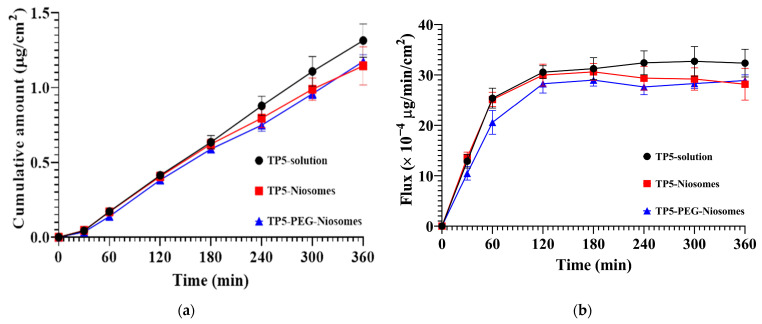
(**a**) Transepithelial transport of cumulative amount of TP5 from TP5 solution, TP5-niosomes, and TP5-PEG-niosomes; (**b**) transepithelial transport of flux rate of TP5 from TP5 solution, TP5-niosomes, and TP5-PEG-niosomes (Mean ± S.D., *n* = 6).

**Figure 14 pharmaceutics-16-00397-f014:**
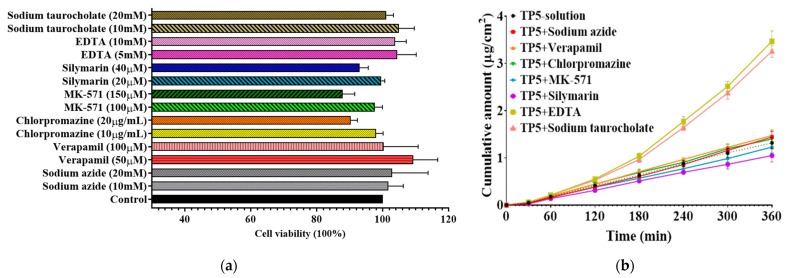
(**a**) Effect of treatment concentrations on the Caco-2 and HT-29 cells’ viability. Cells were incubated with different transporters, inhibitors, and enhancers at 37 °C for 72 h (Mean ± S.D., *n* = 6); (**b**) TP5 cumulative amount in the absence or presence of variable inhibitors and penetration enhancers for 6 h transport study at 37 °C (Mean ± S.D., *n* = 6).

**Figure 15 pharmaceutics-16-00397-f015:**
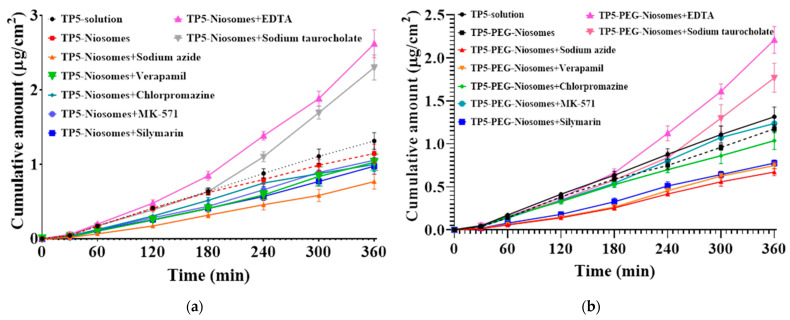
(**a**) TP5 cumulative amount with or without niosomes in the absence or presence of variable inhibitors and penetration enhancers for 6 h transport study at 37 °C (Mean ± S.D., *n* = 6); (**b**) TP5 cumulative amount with or without PEG-niosomes in the absence or presence of variable inhibitors and penetration enhancers for 6 h transport study at 37 °C (Mean ± S.D., *n* = 6).

**Table 1 pharmaceutics-16-00397-t001:** Transporters and inhibitors in Caco-2 and HT-29 co-cultured cell monolayers.

Transporters	Inhibitors	Function	Refs.
Active transport	Sodium azide (10–20 mM)	Adenosine triphosphate (ATP) inhibitor	[26,27]
Clathrin-mediated endocytosis	Chlorpromazine (10–20 μg/mL)	Clathrin-mediated endocytosis efflux inhibitor	[28]
Caveolae-Permeability glycoprotein (P-gp)	Verapamil (50–100 μM)	Caveolae-mediated endocytosis efflux inhibitor	[29,30]
Multidrug resistance-associated protein 2 (MRP2)	MK-571 (100–150 μM)	Protein and peptide transport	[31]
Multidrug resistance-associated protein 5 (MRP5)	Silymarin (20–40 μM)	Protein and peptide transport	[32]
Transport enhancer	EDTA (5–10 mM)	Penetration enhancement	[33]
Transport enhancer	Sodium taurocholate (10–20 mM)	Open tight junction and enhance penetration	[34]

**Table 2 pharmaceutics-16-00397-t002:** The TEER values of Caco-2 and HT-29 cells after 2 weeks with ratios of (a) 1:0, (b) 0:1, (c) 1:1, (d) 1:2, (e) 1:3, (f) 3:1, (g) 5:1, (h) 7:1, and (i) 9:1 (Mean ± S.D. *n* = 6).

Caco-2/HT29 Ratio	TEER Values (Ω cm^2^)
a: 1:0	480.8 ± 45.8
b: 0:1	172.5 ± 18.7
c: 1:1	341.7 ± 37.5
d: 1:2	180.9 ± 21.8
e: 1:3	228.4 ± 28.9
f: 3:1	288.2 ± 22.8
g: 5:1	335.5 ± 36.5
h: 7:1	361.4 ± 39.7
i: 9:1	457.3 ± 40.2

**Table 3 pharmaceutics-16-00397-t003:** The TEER values of Caco-2 and HT-29 cells at 3-day intervals until the TEER values exceeded 350 Ω·cm^2^ (Mean ± S.D. *n* = 6).

Caco-2 and HT-29	Test Day 1	Test Day 2	Test Day 3
TEER (Ω cm^2^)	408.2 ± 36.5	466.8 ± 29.8	485.7 ± 66.8

**Table 4 pharmaceutics-16-00397-t004:** P*app* values of TP5 solution, TP5-niosomes, and TP5-PEG-niosomes from the apical to the basolateral chamber with 6 h transport study (Mean ± S.D., *n* = 6).

Formulations	TP5	TP5-Niosomes	TP5-PEG-Niosomes
P*app* values (10^−8^ cm/s)	5.39 ± 0.46	4.69 ± 0.52	4.81 ± 0.19

**Table 5 pharmaceutics-16-00397-t005:** P*app* values of TP5, TP5-niosomes, and TP5-PEG-niosomes from the apical to the basolateral side after 6 h with different treatments (Mean ± S.D., *n* = 6) (*p* < 0.01 ** to no treatment control).

P*app* Values	Formulations		
(10^−8^ cm/s)	TP5	TP5-Niosomes	TP5-PEG-Niosomes
No treatment	5.39 ± 0.46	4.69 ± 0.52	4.81 ± 0.19
Sodium azide	5.90 ± 0.52	3.15 ± 0.41	2.76 ± 0.18 **
Verapamil	6.03 ± 0.51	4.21 ± 0.45	3.06 ± 0.16 **
Chlorpromazine	5.73 ± 0.63	4.09 ± 0.39	4.25 ± 0.43
MK571	5.04 ± 0.51	4.32 ± 0.51	5.07 ± 0.34
Silymarin	4.30 ± 0.55	4.00 ± 0.45	3.18 ± 0.15
EDTA	14.22 ± 0.89 **	10.73 ± 0.77 **	9.05 ± 0.64 **
Sodium taurocholate	13.36 ± 0.52 **	9.43 ± 0.69 **	7.24 ± 0.69 **

## Data Availability

Data are contained within the article.

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
