# Peer review of "Cellular Uptake and Transport Mechanism Investigations of PEGylated Niosomes for Improving the Oral Delivery of Thymopentin"

_pharmaceutics, 2024, doi:10.3390/pharmaceutics16030397_

Round 1

Reviewer 1 Report

Comments and Suggestions for Authors

pharmaceutics-2875444

Cellular uptake and transport mechanism investigations of PEGylated niosomes for improving the oral bioavailability of Thymopentin

The manuscript by Liu et al. described the development and in vitro evaluation of PEGylated niosomes loaded with Thymopentin. The authors demonstrated that the developed formulation could increase the cellular uptake of the drug. Overall, the manuscript was well organized, and the data were, in part, sufficient for the conclusion. However, it is noted that the formulation was previously developed and published in another article. In this study, the authors presented primarily the cellular uptake of the formulation and its mechanism. The contribution of this work is insufficient for publication as an independent article, particularly in a top-tier journal like Pharmaceutics. Besides, there are other points to consider, as listed below.

1. The title should be changed since there is no in vivo PK study to confirm the improved BA of the formulation.

2. The manuscript has no reference list to evaluate.

3. The total number of references is 87, which is relatively high for an in vitro study like this. Some references may be unnecessary. For example, there is no need for 3 - 7 references to support a statement, like in lines 64, 67, 447, 464, 618, 623, 637, 639, and many others.

4. Lines 50 – 53: Please mention the results of these trials.

5. Citation is required for the statement in lines 74 – 78.

6. Lines 87 – 93: Please correct typos and rewrite this part. Please highlight the novelty and contribution of the study.

7.  Please clarify how to obtain the noisomes numbers. It is inappropriate to use noisome numbers. Instead, please convert to TP5 concentrations.

8. There may be no significant difference in the data comparisons as marked by “*” and “**” in Figure 10. Please recheck.

9. From Figures 10 and 13, there are no improvements in the formulations compared with the free drug.

10. The manuscript can be shortened since all the discussed issues have been well known and available in the literature. The overall findings of this study are limited. This work should be better combined with the next in vivo work in a single article.

Comments on the Quality of English Language

Minor editing of English language required

Author Response

Responding to reviewers concerns/critiques for ‘Cellular uptake and transport mechanism investigations of PEGylated niosomes for improving the oral delivery of Thymopentin’

ID: pharmaceutics-2875444

Thank you for your suggestions and insightful comments. The manuscript has been revised as suggested as showed in below:

  1. Comment 1 from Reviewer 1: The title should be changed since there is no in vivo PK study to confirm the improved BA of the formulation.

  • Response to reviewer: Thanks for your valuable comments. The title has been changed in to ‘Cellular uptake and transport mechanism investigations of PEGylated niosomes for improving the oral delivery of Thymopentin’.

  1. Comment 2 from Reviewer 1: “The manuscript has no reference list to evaluate.

  • Response to reviewer: Thanks for your concern, the references were shown at the end of the manuscript at Line 678, Page 22.

  1. Comment 3 from Reviewer 1: “The total number of references is 87, which is relatively high for an in vitro study like this. Some references may be unnecessary. For example, there is no need for 3 - 7 references to support a statement, like in lines 64, 67, 447, 464, 618, 623, 637, 639, and many others.

  • Response to reviewer: Thanks for your question. We have deleted some references and the total number of references is 68 Please see the updated reference list at Line 678, Page 22.

  1. Comment 4 from Reviewer 1: “Lines 50 – 53: Please mention the results of these trials.

  • Response to reviewer: We have added a brief result of the treatment trails, and please see the updated information at Line 50-54, Page 2.

  1. Comment 5 from Reviewer 1: “Citation is required for the statement in lines 74 – 78. 

  • Response to reviewer: We have added one citation for this paragraph. Please see the updated part at Line 79, Page 2.

  1. Comment 6 from Reviewer 1: “Lines 87 – 93: Please correct typos and rewrite this part. Please highlight the novelty and contribution of the study.

  • Response to reviewer: We have rewrite the paragraph and added the novelty and contribution of this study from Line 88 to 98, Page 3.

  1. Comment 7 from Reviewer 1: “Please clarify how to obtain the niosomes numbers. It is inappropriate to use noisome numbers. Instead, please convert to TP5 concentrations.

  • Response to reviewer: The quantification of niosomes was conducted using NanoSight NS300 (Malvern Instruments Ltd, UK), with variations in niosome numbers achieved through series dilution. Please see the updated part from Line 248 to 252, Page 6.

  1. Comment 8 from Reviewer 1: There may be no significant difference in the data comparisons as marked by “*” and “**” in Figure 10. Please recheck.”

  • Response to reviewer: We have updated Figure 10 with mark “*”. Please see the updated figure at Line 454, Page 15.

  1. Comment 9 from Reviewer 1: “From Figures 10 and 13, there are no improvements in the formulations compared with the free drug.

  • Response to reviewer: TP5 formulations have a similar apparent permeability coefficient compared with the drug itself because TP5 is a small molecule with high solubility. Employing niosomal technology ensures equivalent drug transport rates as free drug solutions while safeguarding the drug from enzymatic degradation in the GI tract. Therefore, there should be no enhancement in the formulations compared with the free drug in the transport studies but a protection effect can be provided for oral delivery. We have added a brief discussion from Line 537 to 542, Page 17.

  1. Comment 10 from Reviewer 1: “The manuscript can be shortened since all the discussed issues have been well known and available in the literature. The overall findings of this study are limited. This work should be better combined with the next in vivo work in a single article.

  • Response to reviewer: We have revised and condensed this manuscript. Please refer to the updated version. Our in vivo investigations, comprising pharmacokinetic and pharmacodynamic assessments, will be addressed alongside formulation design in a separate manuscript. The present paper is dedicated to elucidating the in vitro cell studies.

Reviewer 2 Report

Comments and Suggestions for Authors

Dear authors;

The manuscript is well written and organized, presents some novelty, however before publication some details need to be elucidated:

  1. What challenges do global scientists face regarding thymopentin (TP5)?
  2. How are PEGylated niosomal nanocarriers hypothesized to improve TP5’s properties?
  3. What methods were used to fabricate TP5 loaded PEGylated niosomes?
  4. What types of cells were used as in vitro intestinal models in this study?
  5. What were the results regarding the cytotoxicity of TP5 and its formulations?
  6. How does the cellular uptake of TP5-PEG-niosomes differ from TP5 in solution?
  7. What pathways are involved in the cellular transport of TP5 in solution and its niosomal groups?

Author Response

Responding to reviewers concerns/critiques for ‘Cellular uptake and transport mechanism investigations of PEGylated niosomes for improving the oral delivery of Thymopentin’

ID: pharmaceutics-2875444

Thank you for your suggestions and insightful comments. The manuscript has been revised as suggested as showed in below:

  1. Comment 1 from Reviewer 2: “What challenges do global scientists face regarding thymopentin (TP5)?

  • Response to reviewer: Thanks for your valuable concerns. The challenges of TP5 has been summarized in the introduction part. Please see the highlight paragraph from Line 58 to 67, Page 2.

  1. Comment 2 from Reviewer 2: “How are PEGylated niosomal nanocarriers hypothesized to improve TP5’s properties?

  • Response to reviewer: The Benefits of PEG has been mentioned in the introduction part and we hypothesized that PEGylating a drug delivery system like niosome can improve systemic circulation time and decrease immunogenicity to obtain higher stability of TP5. Please see the highlight paragraph from Line 69 to 75, Page 2.

  1. Comment 3 from Reviewer 2: “What methods were used to fabricate TP5 loaded PEGylated niosomes?

  • Response to reviewer: The fabrication of TP5-PEG-niosomes is based on the conjugation and thin film hydration method. Please see the details of this method from Line 154 to 162, Page 4.

  1. Comment 4 from Reviewer 2: “What types of cells were used as in vitro intestinal models in this study?

  • Response to reviewer: Please see the highlighted section of 2.3.1 Cell co-culture from Line 126 to 148, Page 4. In which, a co-cultured model of Caco-2 or HT-29 cells was selected as the in vitro intestinal models in this study.

  1. Comment 5 from Reviewer 2: “What were the results regarding the cytotoxicity of TP5 and its formulations?

  • Response to reviewer: The results regarding the cytotoxicity of TP5 and its formulations were shown in the 3.1.2 cytotoxicity study from Line 374 to 408, Page 10. In brief, TP5 was not toxic at all the tested concentrations (0 to 10 mg/ml), and PEG-Niosomes with a concentration under 2.00 × 1012 /mL provide a safety profile when treated on human Caco-2 and HT-29 cells.

  1. Comment 6 from Reviewer 2: “How does the cellular uptake of TP5-PEG-niosomes differ from TP5 in solution?

  • Response to reviewer: The results of the cellular uptake of TP5 and its formulations were shown in the 3.1.3 cellular from Line 410 to 510. In summary, cellular uptake of TP5 solution by Caco-2 and HT-29 co-cultured cells are concentration-, time-, temperature- dependent and mainly follow adsorptive-mediated endocytosis and partial passive pathway. The cellular uptake mechanism of TP5-PEG-Niosomes is also concentration-, time-, and temperature- dependent, but follows caveolae or clathrin-mediated endocytosis pathways. TP5-PEG-Niosomes exhibit peak cellular uptake at the 3- or 4-hour mark, whereas TP5 solution reaches its maximum uptake at 2 or 3 hours.

  1. Comment 7 from Reviewer 2: “What pathways are involved in the cellular transport of TP5 in solution and its niosomal groups?

  • Response to reviewer: As mentioned in the results of section 3.1.4 Cellular transport, cellular transport of TP5 mainly through MRP5 endocytosis and passive pathway and effluxed by MRP5 transporters, while TP5-Niosomes and TP5-PEG-Niosomes were through adsorptive- and clathrin- mediated endocytosis with energy participated.

Reviewer 3 Report

Comments and Suggestions for Authors

The paper, "Cellular uptake and transport mechanism investigations of 2 PEGylated niosomes for improving the oral bioavailability of 3 Thymopentin" is exciting and very informative. It would be more interesting if the authors supported the role or literature evidence for EDTA and sodium taurocholate as penetration enhancers. One can find the role of Verapamil, chlorpromazine, filipin, protamine sulfate, etc in cellular uptake or modulating cellular activity.

Author Response

Responding to reviewers concerns/critiques for ‘Cellular uptake and transport mechanism investigations of PEGylated niosomes for improving the oral delivery of Thymopentin’

ID: pharmaceutics-2875444

Thank you for your suggestions and insightful comments. The manuscript has been revised as suggested as showed in below:

  1. Comment 1 from Reviewer 3: “It would be more interesting if the authors supported the role or literature evidence for EDTA and sodium taurocholate as penetration enhancers. One can find the role of Verapamil, chlorpromazine, filipin, protamine sulfate, etc in cellular uptake or modulating cellular activity.

  • Response to reviewer: Thanks for your concerns. Please see the highlighted table from Line 322 to 323, Page 7. In this table, we described the role and cited the literature evidence for different transporters and enhancers.

Round 2

Reviewer 1 Report

Comments and Suggestions for Authors

The manuscript was improved. In some figure legends, only *p<0.05 was mentioned, but ** and *** were used in the figures. Please correct them.

Author Response

We appreciate your attention to detail. The p-values for ** and *** in Figures 5, 10, and 12 have been revised accordingly.

Reviewer 2 Report

Comments and Suggestions for Authors

Dear Authors,

I recommend the publication of the manuscript in the present form, having in account that all the comments of the reviewer were answered. Furthermore the manuscript presents novelty, and is well structured and organized.

Congratulations!

Author Response

We appreciate your constructive feedback and are pleased that you have accepted this manuscript for publication in its current form.